# Gradual Distance Dispersal Shapes the Genetic Structure in an Alpine Grasshopper

**DOI:** 10.3390/genes10080590

**Published:** 2019-08-05

**Authors:** Juan Carlos Illera, Miguel Arenas, Carlos A. López-Sánchez, José Ramón Obeso, Paola Laiolo

**Affiliations:** 1Research Unit of Biodiversity (UMIB, UO-CSIC-PA), Oviedo University, 33600 Mieres, Spain; 2Department of Biochemistry, Genetics and Immunology, University of Vigo, 36310 Vigo, Spain; 3Biomedical Research Center (CINBIO), University of Vigo, 36310 Vigo, Spain; 4Department of Biology, Organisms and Systems, GIS-Forest Group, Oviedo University, 33600 Mieres, Spain

**Keywords:** Cantabrian Mountains, *Chorthippus cazurroi*, coalescent simulations, migration models, sky islands, incipient diversification, flightless grasshopper

## Abstract

The location of the high mountains of southern Europe has been crucial in the phylogeography of most European species, but how extrinsic (topography of sky islands) and intrinsic features (dispersal dynamics) have interacted to shape the genetic structure in alpine restricted species is still poorly known. Here we investigated the mechanisms explaining the colonisation of Cantabrian sky islands in an endemic flightless grasshopper. We scrutinised the maternal genetic variability and haplotype structure, and we evaluated the fitting of two migration models to understand the extant genetic structure in these populations: Long-distance dispersal (LDD) and gradual distance dispersal (GDD). We found that GDD fits the real data better than the LDD model, with an onset of the expansion matching postglacial expansions after the retreat of the ice sheets. Our findings suggest a scenario with small carrying capacity, migration rates, and population growth rates, being compatible with a slow dispersal process. The gradual expansion process along the Cantabrian sky islands found here seems to be conditioned by the suitability of habitats and the presence of alpine corridors. Our findings shed light on our understanding about how organisms which have adapted to live in alpine habitats with limited dispersal abilities have faced new and suitable environmental conditions.

## 1. Introduction

Understanding the evolution of Pleistocene glacial periods in temperate regions has been instrumental in disentangling much of the biogeographic distributions and genetic structure of their biotas. The effect of Quaternary climatic oscillations on the extant genetic diversity in the European and North American biotas is well-documented [1,2,3,4]. The role of the southern European peninsulas acting as glacial refugia for terrestrial biotas during glacial periods, and serving as the cradle for postglacial expansions after the retreat of the ice sheets, has been confirmed in Mediterranean, continental and alpine species [5,6,7,8]. Among these groups, alpine taxa have been most strongly shaped in their evolutionary history by the geographic location and complexity of the European high mountains, and a large number of taxa are now endemic to these mountain ranges [9,10,11]. However, we are far from a detailed understanding about the genetic consequences of shifting distributions in taxa with restricted dispersal abilities, and especially on species living along the sky islands of southern Europe.

Mountain chains are considered hot spots of biodiversity due to the high levels of unique taxa they host, which is a direct consequence of harbouring a rich range of habitats in a relatively small area [12,13,14]. Biotas occurring on mountains are well known for developing unique adaptations in physiology, morphology, and life histories for coping with extreme environmental conditions [15,16]. One of the strategies that repeatedly occurs in ground dwelling invertebrate taxa inhabiting alpine habitats is the loss of flight [17]. This adaptation is explained due to the existence of conflicting selection pressures. Thus, the costs saved from the development of flight traits could enhance fat reserves available for reproduction and self-maintenance [18,19]. However, the adaptation to become flightless also provides handicaps such as limited responses to cope with environmental changes due to their poor dispersal abilities. On the other hand, mountains do not form continuous lines of habitat where species occur everywhere without restrictions, but the presence of unsuitable habitats such as valleys and rivers surrounding the mountains produce discontinuities on their distributions. Such a mosaic of suitable and unsuitable landscapes can create barriers to gene flow for neighbouring populations restricted to mountaintops [20], and eventually promote isolation and differentiation over small spatial scales [21,22].

Understanding how flightless organisms cope with environmental changes can inform us about how heterogeneous landscapes shape the genetic structure of montane biotas over time. Ultimately, this information will shed light on functional connectivity of disjunct sky islands, providing robust guidelines for conservation managers. Under a recurrent scenario of glacial and interglacial periods, it is not evident how animals with reduced dispersal capabilities respond, although it must be also dependent on the speed of the environmental changes [23]. It is plausible to speculate that during glacial periods populations use glacial refuges around mountains until climatic conditions are suitable to dispersal [24]. However, understanding how species follow the progressive retreat of glaciers is not trivial because the species could spend long periods on their glacial refuges before starting post-glacial dispersal events. In addition, extant distributions could be the result of both long-distance dispersal (LDD) and gradual-distance dispersal (GDD) events, and these models are rarely evaluated together [25]. LDD in small flightless organisms may be mediated by upland winds [26], or through post-regurgitation events mediated by birds [27,28]. In the last scenario, grasshoppers could survive after digestion and, therefore, colonise regions far away. GDD is the expected model for non-vagile animals since it implies a progressive occupancy of territory through short-distance dispersal events [29]. Discriminating between migration through LDD and GDD is crucial in improving our understanding of the dynamics and evolution of the ecological system, and it is also an important challenge for biodiversity conservation. In particular, this information can provide robust guidelines to predict changes in population growth and distribution range under future scenarios of global change. Ideally, such findings should provide a set of conservation priorities with cost-effective strategies in order to preserve alpine biodiversity over time [30,31].

With the present study we aimed at unravelling the mechanisms explaining the dispersal process and evolution of spatial genetic structure in an alpine flightless grasshopper (*Chorthippus cazurroi*). This grasshopper shows a patchy distribution in the Cantabrian Mountains, occurring along a narrow range between 1400 and 2570 m above sea level. Such a narrow distribution appears ideal to make inferences about how landscape features shaped the demographic history and genetic variability of alpine taxa. To achieve our objectives, firstly, we scrutinised the maternal genetic structure from all Cantabrian sky islands where this taxon occurs. We studied mitochondrial data because when the range of a species is exhaustively sampled, it is possible to get robust inferences of its genetic structure due to the absence of recombination and fast coalescent time [32,33]. Next, we evaluated the fitting of two migration models (LDD and GDD) to explain genetic structure of the studied populations. To do so we applied approximate Bayesian computation (ABC) approaches based on extensive spatially explicit computer simulations. Finally, we estimated population genetic parameters assuming the best-fitting migration model.

## 2. Material and Methods

### 2.1. Species and Sampling

*Chorthippus cazurroi* is an endemic grasshopper of the Cantabrian Mountains (Figure 1). The very short length of their wings makes this species incapable of flight, which has been confirmed with our own observations in situ. Thus, *Chorthippus cazurroi* shows a wing length of 8.05 ± 0.86 mm (mean ± Standard Deviation) for females (*n* = 285) and 8.27 ± 0.74 mm for males (*n* = 156), meanwhile, *C. yersini* (a relative species capable of short flight and occurring in sympatry) shows longer wing length: 22.07 ± 1.46 mm and 17.88 ± 1.42 mm for females (*n* = 189) and males (*n* = 134), respectively. However, both species show closer total body length (19.01 ± 1.83 mm versus 27.20 ± 1.46 mm) for *C. cazurroi* and *C. yersini* females, respectively; and 13.53 ± 1.63 mm versus 20.34 ± 1.56 mm for males, respectively. The species is found on grasslands habitats over 1400 m above sea level (a.s.l.) on limestone massifs. This grasshopper species shows the typical univoltine life cycle with nymphs emerging after snow is melted (June–July). After four moults, the nymphs mature to adult stage, which appears to be especially abundant in August, when most reproduction occurs. In controlled laboratory conditions, with food provided ad libitum, females survive on average 21 ± 1 Standard Error days, during which they can lay up to 11 egg pods, each containing on average 8 ± 3 SE eggs [34].

We sampled all populations of *Chorthippus cazurroi* in the Cantabrian Mountains. In particular, the species was found in 22 (Table 1, Figure 1) out of 189 sampled localities [35]. We surveyed an area of 10,000 km^2^ from the sea level to mountain-tops during 2012–2015 and the species was found in an area of 380 km^2^ [35,36,37]. When a grasshopper population was localised, we caught individuals by hand or using a sweep net (if many individuals were detected) with a maximum of 25 specimens per population. All individuals sampled were stored alive in 50 mL centrifuge tubes until they were frozen in the lab at −20 °C or −45 °C. With such sampling effort, we are confident that we included individuals from all sky islands where this endemic grasshopper occurs.

### 2.2. DNA Extraction and Sequencing

We extracted DNA from the hind femur or head using the Qiagen DNeasy kit (Venlo, Netherlands) and following the manufacturer’s protocol. To determine genetic variability of *Chorthippus cazurroi*, we amplified a 575 bp fragment of the cytochrome oxidase subunit I (COI) using the primers LCO1490 and HCO2198 [38]. Polymerase chain reactions (PCR) were set up in 10 µl total volumes following the procedures and conditions described in previous studies [36,39]. PCR products were checked by electrophoresis on 1.5% agarose gels stained with GelRed^TM^ nucleic acid gel stain (Biotium, Inc., Hayward, CA, USA). We performed sequencing reactions using the Perkin Elmer BigDye v. 3.1 (Applied Biosystems, Carlsbad, CA, USA) terminator reaction mix in a volume of 10 µL, following the procedures and conditions described in Pato et al. [39]. The final product was purified and sequenced on an ABI PRISM^®^ 3130xl Genetic Analyzer (Thermo Fisher Scientific Inc., Waltham, MA, USA). Sequences were edited and aligned by hand using BioEdit v. 7.2.5 [40]. Unique sequences were deposited in the GenBank database (NCBI) with the accession numbers MH271324–MH271356.

### 2.3. Estimation of Genetic Diversity

We calculated genetic diversity statistics (number of haplotypes, haplotype and nucleotide diversity [41], and theta (2N_e_µ) [42] per locality, using the program DnaSP version 5.10.01 [43]. We inferred relationships among populations performing an unrooted phylogenetic network analysis using the program TCS v. 1.21 [44].

### 2.4. Migration Model Selection and Estimation of Population Genetics Parameters

#### 2.4.1. Spatially-Explicit Computer Simulations

We evaluated how two migration models, LDD and GDD (details about these migration models are shown in the Appendix A), fit the real data. For every evolutionary model we performed a total of 30,000 spatially-explicit computer simulations with the program SPLATCHE3 [45] that implements a variety of population and environmental capabilities including LDD [46]. We performed 100 additional simulations, hereafter called as pseudo-observed data sets (PODS), for every model to evaluate the statistical power of model selection and parameter estimation. An overall description of the models implemented in SPLATCHE3 is shown in Appendix A (for further details see [45]).

We used a landscape composed of 46,292 squared demes (subpopulations) of 0.1 km^2^ each (see Figure 1 and Appendix A) that covers all the demes above 1400 m, considering that below this altitude *Chorthippus cazurroi* individuals were not found. We assumed that the range expansion started from a single deme (Urriellu, Figure 1 and Figure 2) based on the combination of high values of genetic diversity found on this locality (Appendix A) and its close situation to putative areas with potential habitat for grasshoppers during the Pleistocene [47]. We simulated population genetics parameters by sampling from prior distributions defined according to previous studies (see Appendix A). The simulated range expansion started with the retreat of the ice sheets derived from the last glacial maximum period (LGM), and thus it followed a prior distribution in between 15–20 kya [48]. We considered a prior distribution for the ancestral population size ranging between 100–1000 individuals and a population growth rate varying within 0.2–0.9. We also considered a prior distribution for the migration rate varying among 5% and 30% of migrants per generation according to previous studies (Appendix A). For the model with LDD, we applied a proportion of migrants with LDD ranging from 0.001 to 0.05 (proportion of individuals experiencing an LDD event), in agreement with Alves et al. [25]. Then, we assumed parameters of the gamma distribution derived from other terrestrial species, in absence of available information for closer taxonomic units [46,49]. We arbitrarily set the maximum distance of an LDD event to 100 demes (10 km) in order to avoid unrealistically long LDD events and a minimum of LDD distance to more than 1 deme (>100 m). DNA sequences were simulated under a prior distribution for the mutation rate that followed Shapiro et al. [50].

#### 2.4.2. Summary Statistics

We grouped the 22 localities into seven regions according their spatial distribution (Appendix A and Table 1). Summary statistics (SS) were computed with Arlequin version 3.5 [51] for the simulated and real data. We constrained our ABC analysis to only biologically informative SS. Thus, we did not apply partial least squares (PLS) components [52,53] to avoid their lack of biological meaning (in comparison to genetic SS such as pairwise differences or Fixation index (FST), and because PLS components computed separately present a different meaning (i.e., complicating comparisons of PLS components obtained from different data). Altogether, we applied 31 SS: (i) Genetic diversity in every group and in all groups together (including pairwise differences, π and their standard deviation), and (ii) genetic differentiation (FST) among every combination of two groups and from all groups together. We found that this set of SS provides robust ABC inferences (see Results).

#### 2.4.3. Selection among Migration Models

As indicated above, we generated 30,000 simulated datasets for each of the two migration models. In addition, we simulated 100 PODS for each model to evaluate the accuracy of the selection method. Considering that the accuracy of the estimation for a particular real dataset may differ among ABC techniques [54,55], we applied four different ABC approaches to perform the selection of the best-fitting evolutionary model. The four ABC approaches are as follows: The rejection approach developed by Pritchard et al. [56] (Pr), and the rejection (Rrej), regression (Rreg) and neuralnet (Rnn) approaches implemented in the abc package of R [57,58]. The power of the Pr approach was evaluated with the 100 PODS, whilst the power of the remaining approaches was evaluated with the leave-one-out cross-validation method implemented in the abc library of R (function cv4postpr) that accounts for nonlinearity adjustment based on 100 permutations [58]. We assumed a tolerance of 0.1 according to the authors’ recommendation [58], thus retaining the 3000 simulations with SS closest to the SS of the real data. The model selection based on the fitting with the real dataset was performed with all the approaches indicated above.

#### 2.4.4. Estimation of Population Genetic and Environmental Parameters

We estimated the population genetic parameters with the Rrej, Rreg, and Rnn ABC approaches [58]. We used the model that best fitted the real data, and again we retained the 3000 simulations closest to the real data. First, we investigated the power of the parameters estimation with the 100 PODS by computing for each parameter the distance between the true (simulated) value of the POD and the estimated value. Finally, we performed the parameter estimation from the real dataset with the three ABC approaches previously indicated.

## 3. Results

### 3.1. Genetic Diversity

*Chorthippus cazurroi*’s whole distribution range included 22 localities along the sky islands of the Cantabrian Mountains (Figure 1 and Appendix A). In total, we sampled 327 individuals showing an incipient process of differentiation with 33 unique haplotypes found. The haplotype network shows a central haplotype shared by five out of the seven groups analysed, plus other two haplotypes also shared by other two groups (Appendix A). Finally, the highest maternal genetic variability was found in the eastern and western populations (Appendix A).

### 3.2. Selection of the Best-Fitting Migration Model

We found a high probability (above 0.8) for distinguishing between the investigated models (LDD versus GDD) under all the applied ABC approaches (Appendix A). The analysis of the real data shows that the GDD model fits the real data with much higher posterior probabilities (above 0.85) than the LDD model, independently of the ABC approach applied (Table 2).

### 3.3. Estimation of Population Genetics and Environmental Parameters

We found that the ABC approaches provided a reasonable power to estimate the population genetics parameters under the GDD model, after using the 100 PODS, although, as expected, the accuracy of this estimation slightly varied among the studied parameters and the applied ABC approaches (Appendix A). The population genetics parameters estimated with all the ABC approaches under the GDD model are shown in Appendix A, and in Table 3 (presented considering the estimation power of the applied ABC approaches). The results show a time of the onset of the expansion with a mode around 19,300 years, related to the end of the LGM, and an ancestral population size around 640 (540–706) individuals. We also found a small population growth rate of 0.28 (0.27–0.29), a low migration rate of 0.12 (0.06–0.27) and a regular carrying capacity of 177 (105–313). Finally, the estimated mutation rate was 1.37 10^-06^ (5.15 × 10^−7^–9.43 × 10^−6^). Here we present the parameter values estimated with the highest probability (mode of the posterior distribution), and the variance in the precision of these estimates (we show the 90% Highest Posterior Density Interval—HPDI) for the interpretation of these results.

## 4. Discussion

Our inferred haplotype network suggests a scenario of recent colonisation of the Cantabrian Mountains sky islands, led by a starting population of more than 500 individuals. Our findings support a colonisation by a gradual dispersal process rather than the presence of LDD events such as those caused by upland winds or biotic factors. Such a contemporary expansion is well-matched with the limited genetic structure found, which explains the pattern of incomplete lineage sorting revealed by the haplotype network analysis, and presenting haplotypes shared between massifs. This genetic structure is compatible with a scenario of incomplete lineage sorting, which is common in species with recent differentiation processes [59,60].

The end of the last glacial period, and the posterior retreat of the ice sheets in the Cantabrian Mountains, provided new and suitable alpine landscapes dominated by dispersal shrubs and grasses [61]. In addition, human management during Holocene has limited the tree line below 1600 m a.s.l., favouring the expansion and persistence of grassland and scrubland habitats above this altitude [62]. Our results suggest that after glacial retreat, *Chorthippus cazurroi* started a process of slow and gradual range expansion along the Cantabrian sky islands using, most likely, narrow alpine corridors to achieve its current distribution (Figure 2). Our findings also show a slight genetic differentiation of new founded populations, probably due to local adaptations and genetic drift [36,39]. Interestingly, our findings suggest a scenario where gene flow could be mostly determined by the proximity of neighbouring populations, which agrees with the pattern of isolation by distance found in this species [39].

Mountains are often compared with oceanic islands due to their characterisation as discrete entities, sharing a high number of endemic species along relatively small areas. In addition, both systems have been dramatically affected by Pleistocene glaciations resulting in significant shifts in spatial and temporal species distribution and population genetic structure [63,64,65]. The legacy of climatic oscillations has also fostered local extinctions, loss of connections between populations formerly connected and driven diversification episodes in terrestrial biotas [4,20,66,67]. Recent studies performed with flightless and winged alpine grasshoppers have provided evidence of colonisation by populations located on glacial refuges during the Pleistocene glaciations [10,63]. Hence, recurrent episodes of range expansion and contraction driven by climatic fluctuations likely shaped the genetic divergence observed by drift and local adaptation mechanisms [68]. It also seems plausible to speculate that such a scenario of isolation, drift, local adaptation and introgression events promoted diversification and shaped the genetic differentiation recorded in alpine biotas [10,11,24,69]. Our findings showed a limited genetic differentiation within *Chorthippus cazurroi* populations, which is compatible with a pattern of recent dispersal throughout the Cantabrian Mountains. Such a pattern agrees with our knowledge about the magnitude and distribution of glacial sheets in the Cantabrian Mountains, which were persistent, continuous, and predated the ages reported at other massifs such as Pyrenees and the Italian Apennines [48].

On balance, the specific climatic oscillations experienced by these northern massifs could have conditioned the colonisation and occurrence of this flightless grasshopper along the Cantabrian sky islands. Interestingly, our results show how this mountain grasshopper has negotiated through narrow corridors within the unsuitable landscapes (i.e., valleys, woodlands and rivers) surrounding the sky islands. This finding is relevant in order to predict future movements of this species and assess ecological connectivity among disjunct sky islands distributed throughout heterogeneous and unsuitable landscapes. Undoubtedly, this information will be also critical in future management actions with the goal of preserving genetic diversity and ecosystem function. Importantly, our results confirm that this flightless species do not participate in LDD movements or, if they did, such circumstance did not shape its current genetic structure. Long distance dispersal constitutes a recognised strategy performed by diverse organisms with the aim of colonising areas away from the source, rapidly, and without loss of genetic diversity [25,29,70]. Indeed, LDD allows the maintenance of genetic diversity in fragmented habitats [71]. However, our results support a scenario of gradual migration with a moderate proportion of individuals migrating between neighbouring demes. We also found low population growth rates and carrying capacity, suggesting a small increase of population size over time. These values seem low in relation to other grasshopper communities studied [72]. Nevertheless, conversely to other grasshopper communities, our studied populations inhabit alpine habitats characterised by harsh environmental conditions and presence of nutrient-poor plant communities [37], which could limit both growth rate and carrying capacity. Finally, the estimated mutation rate was in agreement with previous studies [73], although this parameter may change among studies, especially when is estimated from different molecular markers.

Our approach provides a framework to improve our understanding, and ecological meaningfulness, of colonisation and diversification processes of non-vagile organisms throughout the sky islands of Cantabrian Mountains. Nonetheless, we are aware that our biologically informed hypotheses hinge upon required assumptions, which can be challenging. Our models assume an ancestral population from which the species spread along the sky islands. We chose such a source population according to our knowledge on the evolution of the last glacial period in the Cantabrian Mountains, the potential habitats where the ancestral population could have occurred, and the high values of genetic diversity found. However, the clear pattern of migration shown by our analyses (GDD) is unlikely to be affected by the origin of the expansion [74]. Another hypothesis consists of several origins of dispersal (instead of only one) along the Cantabrian Mountains. According to this alternative hypothesis, during the LGM, grasshopper populations had been connected at lower altitude localities (i.e., below the glaciers). After glacial retreat, individuals from the grasshopper populations started a slow colonisation process of sky islands with posterior geographical isolation of these populations. However, based on the restricted range of this species, the limited genetic differentiation found and the spatial variation of genetic diversity, we considered more parsimonious a gradual colonisation starting from a single source. Future studies involving higher genomic coverage are now needed to evaluate and discriminate between these hypotheses on the origin of dispersal. Another assumption made by our models is the lack of variation of population genetics parameters (i.e., population growth rate, migration rate and carrying capacity) over time and space, which are necessary to avoid over-parameterization and allow the estimation of parameters with certainty.

Overall, we showed that *Chorthippus cazurroi* populations established in the Cantabrian Mountains present a simple pattern of progressive and slow gradual migration. We also found that these populations evolved with a slow increase of their population sizes over time, which could be explained by the patchy distribution of suitable habitats and hard environmental conditions present in the region. In conclusion, our findings have shed light on the processes involved in the formation of biodiversity in alpine flightless animals occurring on southern temperate mountains and, importantly, have provided an ideal framework to integrate genetic divergence at the population level with ecological information related to animal differentiation such as morphological and acoustic traits.

## Figures and Tables

**Figure 1 genes-10-00590-f001:**
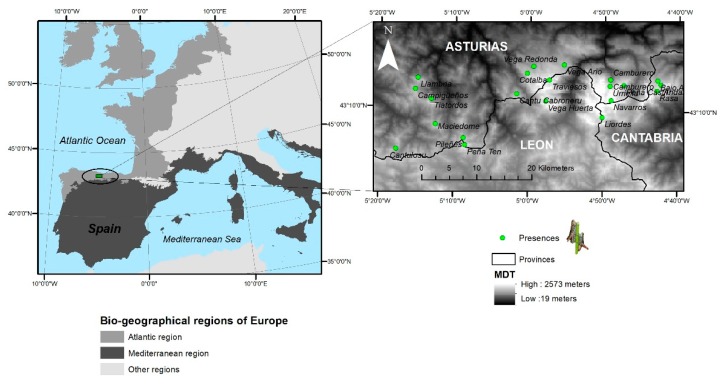
Distribution of *Chorthippus cazurroi* along the sky islands of the Cantabrian Mountains. Localities with individuals sampled for this study are depicted with green dots.

**Figure 2 genes-10-00590-f002:**
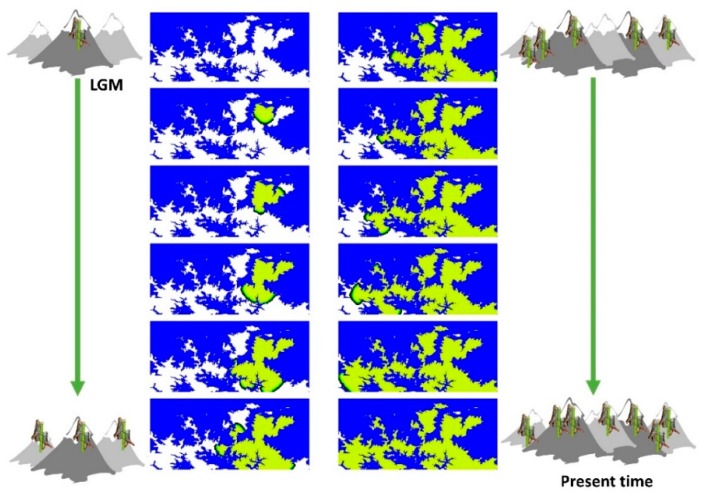
Illustrative example of spatially-explicit computer simulations performed with SPLATCHE3 under the gradual distance dispersal (GDD) model. The landscape corresponds to a grid of demes with size 0.1 km^2^ where populations can only live in demes above 1400 m. Demes that cannot be colonised are shown in blue, empty demes (uncolonised) are shown in white, and colonised demes are shown in green. The presented snapshots were collected every 50 generations and mimic the colonisation of the area after the last glacial period (LGM). Mountain and grasshopper icons depict increasing geographical distribution and numbers over time.

**Table 1 genes-10-00590-t001:** Sampled localities, with their acronyms and sample size, analysed in this study (see Figure 1 for a visualisation of their geographic distribution). “Group” indicates the assignation of localities into groups according to their geographic distribution (Appendix A).

Locality	Acronym	Sample size	Group	Massif
Cantu l’Osu	COsu	20	1	Central
Campigüeños	Cam	11	2	Western
Llambria	Lla	11	2	Western
Tiatordos	Tia	16	2	Western
Maciédome	Mac	18	2	Western
Peña Ten	Ten	15	3	Western
Pileñes	Pil	10	3	Western
Cantu Cabroneru	CC	15	4	Central
Traviesos	Tra	15	5	Central
Cotalba	Cot	15	5	Central
Vega Ario	Va	15	5	Central
Vega Huerta	Vh	16	5	Central
Vegarredonda	VR	17	5	Central
Tiros Navarros	NV	25	6	Eastern
Peña Castil	PC	16	6	Eastern
Liordes	Lio	6	6	Eastern
Urriellu	U	19	6	Eastern
Camburero	Camb	3	6	Eastern
Morra Lechugales	MoHie	19	7	Eastern
Andara	A	15	7	Eastern
Casetón Andara	Ba	15	7	Eastern
Rasa	Ras	15	7	Eastern

**Table 2 genes-10-00590-t002:** Fitting of the GDD and long-distance dispersal (LDD) migration models with the real data using four approximate Bayesian computation (ABC) approaches. The GDD migration model was the best fitting model (probabilities above 0.85 under any applied estimation approach). The ABC estimation approaches are as follow: Pr, Pritchard’s rejection approach; Rrej: rejection approach implemented in the abc library of R; Rreg, multiple regression approach implemented in the abc library of R; Rnn, neuralnet method rejection approach implemented in the abc library of R.

Model	GDD				LDD			
**ABC approach**	Pr	Rrej	Rreg	Rnn	Pr	Rrej	Rreg	Rnn
**Probability**	0.97	0.87	1.00	1.00	0.03	0.13	0.00	0.00

**Table 3 genes-10-00590-t003:** Estimates of population genetic parameters from the real data under the best fitting migration model and considering the most accurate ABC approach to estimate each parameter. The estimations were performed with the rejection (Rrej), regression (Rreg) and neuralnet (Rnn) approaches implemented in the abc library of R [57]. The estimates (mode, mean or median) with the higher accuracy (see Appendix A) are shown in bold.

Parameter	ABC Approach	Mode	Mean	Median	90% HPDI ^1^
Time of the onset of the expansion ^2^	Rejection	**19,336**	17,547	17,558	15,256–19,775
Ancestral population size	Neuralnet	662	**638**	649	540–706
Population growth rate	Regression	**0.28**	0.28	0.28	0.27–0.29
Migration rate	Rejection	0.07	0.14	**0.12**	0.06–0.27
Carrying capacity	Rejection	122	**177**	154	105–313
Mutation rate	Rejection	**1.37 × 10^−6^**	4.78 × 10^−6^	4.66 × 10^−6^	5.15 ×10^−7^–9.43 × 10^−6^

^1^ 90% HPDI indicates the 90% highest posterior density interval. ^2^ Times shown in years and generations (generation time is 1 year).

## Data Availability

Sequences were deposited in the GenBank database (NCBI) with the accession numbers MH271324-MH271356.

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
