# Peer review of "Gradual Distance Dispersal Shapes the Genetic Structure in an Alpine Grasshopper"

_genes, 2019, doi:10.3390/genes10080590_

Round 1

Reviewer 1 Report

According to me the flishtless of the sepcies should be better explained and demonstrated. All the discussion is based on this and so it could be better to investigate also the wing morphology. I suggest also to add the other species of Orthoptera present ion the area telling if they are fliyng or not....

Author Response

According to me the flishtless of the sepcies should be better explained and demonstrated. All the discussion is based on this and so it could be better to investigate also the wing morphology. I suggest also to add the other species of Orthoptera present ion the area telling if they are fliyng or not....

>> We have provided further details on the morphology in the main text. The very short length of their wings make this species incapable of flight, (such a circumstance is very easy to confirm in the field). We have provided own specific measurements of wing length and total body length taken both Chorthippus cazurroi and another related species (Chorthippus yersini) occurring in sympatry, which is capable of short flight. Looking at these measurements it is clear how the length wing of C. cazurroi is extremely shorter than C. yersini despite they show similar total body length. We provide pictures of C. cazurroi for your consideration as well.

Now, it reads (lines 97-104):

“The very short length of their wings makes this species incapable of flight, which has been confirmed with our own observations in situ. Thus, Chorthippus cazurroi shows a wing length of 8.05 ± 0.86 mm (mean ± SD) for females (n = 285) and 8.27 ± 0.74 mm for males (n = 156), meanwhile, C. yersini (a relative species capable of short flight and occurring in sympatry) shows longer wing length: 22.07 ± 1.46 mm and 17.88 ± 1.42 mm for females (n = 189) and males (n = 134), respectively. However, both species show closer total body length (19.01 ± 1.83 mm versus 27.20 ± 1.46 mm) for C. cazurroi and C. yersini females, respectively; and 13.53 ± 1.63 mm versus 20.34 ± 1.56 mm for males, respectively.

The reviewer asks for adding the name of the grasshopper in the abstract and keywords.

>> We have added the Latin name in the keywords. However, we have not followed such suggestion in the abstract due to the strict number of words allowed.

Line 37. The reviewer asks to add some old references.

>> We have followed his/her suggestion and we have added this one [4]:

La Greca. 1954. Le Cophopodisma (Orth. Catantopidae) dell'Appennino ed il loro differenziamento infraspecifico. Annuario dell'Istituto e Museo di zoologia dell'Università di Napoli. 6(7):1-20

We have updated the list of references along the manuscript too.

Line 42. large number is not very precise. It could be better to express the number of European species (taxa even better) and the high mountains ones.

>> We do not think necessary to provide a specific number of species in the main text, especially because we are supporting our statement with three references. However, we agree that is more suitable to talk about taxa instead species, and we have changed such a word. Line 42.

Lines 50-51. The flightless of Chortippus cazurroi is not clearly demonstrated. many shorthened wings specis can fligh and it could be good to show a specimen with tegnmina and wings opened. In addition several sepcies exclusive to high mountains fligh.

>> See our answer in point (1). Briefly, we have provided morphological and observational details showing that C. cazurroi is a flightless species.

Sure? Wind is more probable and not only probable...

>> There are previous examples showing dispersal through post-regurgitation events mediates by birds, therefore, it is suitable to introduce this hypothesis here.

Is it know in many years the eggs etch? In many alpline orthoptera and noty only alpine, they etch in different years.

>> We are providing own information obtained in laboratory conditions.

In Italian Italopodisma was showed that several glacial periods and relative warmer periods generate recombination of populations and even speciation....

>> Our results support that in our species started the dispersal and differentiation process in the Cantabrian Mountains at the end of the last glacial period.

Lines 268-269. See the works of La Greca too.

>> Ok, we have added one of these studies [4]

Reviewer 2 Report

This is an interesting, well designed and written phylogeographic study which analysed mechanism explaining the colonisation of mountains (sky islands) by an endemic and flightless grasshopper species. In order to accept authors conclusions unconditionally, I have only one single reservation (=major comment) concerning the number of molecular markers. It is a rather uncommon today to employ single mtDNA marker only for such inference. On the other hand, I am impressed how much authors could harvest from this relatively poor genetic information. I would thus suggest to argue a bit more about adequacy of used approach and to discuss robustness of such limited results. 

Minor comments

Fig. 1. Please make larger detailed map with sites comparing to Europe which could be arranged as some inset there.

Fig. 2. It is not clear what mountain icons represent there. Was the mountain area smaller during LGM? Grasshoppers icons are really hard to distinguish.

Table 3. Nothing is shown in bold there.

Author Response

This is an interesting, well designed and written phylogeographic study which analysed mechanism explaining the colonisation of mountains (sky islands) by an endemic and flightless grasshopper species. In order to accept authors conclusions unconditionally, I have only one single reservation (=major comment) concerning the number of molecular markers. It is a rather uncommon today to employ single mtDNA marker only for such inference. On the other hand, I am impressed how much authors could harvest from this relatively poor genetic information. I would thus suggest to argue a bit more about adequacy of used approach and to discuss robustness of such limited results.

>> We thank the reviewer for the positive comments about our study.
We acknowledge that our data is limited in terms of sequence length. However, we have analysed 327 individuals from 22 geographically disjunct populations, and the evaluation of our analyses (both model selection and parameters estimation) showed that this amount of information was sufficient to yield acceptable estimates. In the first version, we already explained (lines 88-90) the advantages of studying mitochondrial data when the range of a species is exhaustively sampled (such as occurred in our case). Finally, we have indicated that “future studies involving higher genomic coverage are now needed to evaluate and discriminate between these hypotheses on the origin of dispersal“ (lines 324-326).

Minor comments

Fig. 1. Please make larger detailed map with sites comparing to Europe which could be arranged as some inset there.

>> We do not understand what reviewer is asking us. With this map, we are showing the world distribution of our endemic species in relation with the Iberian Peninsula and part or Europe. In addition, we are providing the geographical position in degrees and minutes, and the scale (km). In the case we would add a larger portion of Europe in our maps, we would decrease the accuracy of the distribution of our species, which is very restricted.

Fig. 2. It is not clear what mountain icons represent there. Was the mountain area smaller during LGM? Grasshoppers icons are really hard to distinguish.

>> Mountain and grasshoppers icons represent how grasshoppers are increasing in number and how they are spreading through the Cantabrian Mountains. We have explained the meaning of icons in the legend now.

“Mountain and grasshopper icons depict increasing geographical distribution and numbers over time.”

Table 3. Nothing is shown in bold there.

>> We have added the bold style there. In addition, we noticed that the last part of the legend (just below of Table 3) was missed. We have added that too.